# An Overview on the Associations between Health Behaviors and Brain Health in Children and Adolescents with Special Reference to Diet Quality

**DOI:** 10.3390/ijerph17030953

**Published:** 2020-02-04

**Authors:** Sehrish Naveed, Timo Lakka, Eero A. Haapala

**Affiliations:** 1Physiology, Institute of Biomedicine, University of Eastern Finland, Kuopio Campus, 70211 Kuopio, Finland; Timo.Lakka@uef.fi (T.L.); Eero.a.haapala@jyu.fi (E.A.H.); 2Department of Clinical Physiology and Nuclear Medicine, Kuopio University Hospital and University of Eastern Finland, 70211 Kuopio, Finland; 3Kuopio Research Institute of Exercise Medicine, 70100 Kuopio, Finland; 4Faculty of Sport and Health Sciences, University of Jyväskylä, 40014 Jyväskylä, Finland

**Keywords:** health behaviors, lifestyles, diet quality, nutrition, brain, brain health, cognition, learning, children, adolescents

## Abstract

Unhealthy diet has been associated with overweight, obesity, increased cardiometabolic risk, and recently, to impaired cognition and academic performance. The aim of this review is to provide an overview of the associations between health behaviors and cognition and academic achievement in children and adolescents under 18 years of age with a special reference to diet quality. Dietary patterns with a low consumption of fish, fruits, and vegetables, and high in fast food, sausages, and soft drinks have been linked to poor cognition and academic achievement. The studies on the associations between the high intake of saturated fat and red meat and low intake of fiber and high-fiber grain products with cognition are limited. The available evidence and physiological mechanisms suggest that diet may have direct, indirect, and synergistic effects on brain and cognition with physical activity, sedentary behaviors, cardiometabolic health, and sleep, but the associations have been modest. Therefore, integrating a healthy diet, physically active lifestyle, and adequate sleep may provide optimal circumstances for brain development and learning. We conclude that most of the existing literature is contained in cross-sectional studies, which therefore highlights the need for longitudinal and intervention studies on the effects of diet, physical activity, sedentary behavior, and sleep on cognition and academic performance.

## 1. Introduction

Poor nutrition may impair rapidly developing brain and cognitive functions, and low quality diets may also deteriorate the academic achievement of children [1]. Poor cognition and academic achievement in childhood have been linked to an increased risk of adulthood obesity, unemployment, and low socioeconomic positioning in adulthood, suggesting that it is important to identify possibilities to support brain development, cognition, and academic achievement in childhood [2,3,4]. Given that proper nutrition is important for brain development, cognition, and academic achievement, recent evidence concerning the dietary patterns of children is alarming. A child’s diet typically includes a high intake of saturated fat and sucrose, a high consumption of fast foods, and a low consumption of vegetables [5,6,7,8,9].

There is also increasing evidence that children and adolescents grow physically inactive and spend most of their waking hours being sedentary [10]. It has been estimated that approximately half of children achieve the recommended 60 min of moderate-to-vigorous physical activity and the proportion decreases as age increases [11,12]. Some studies have also provided evidence that sleep duration has decreased during the past decades in children and adolescents [13,14]. Increased use of electronic devices (e.g., computers and mobile phones) in the evening may also compromise sleep [15].

The prevalence of pediatric overweight has also increased dramatically during the past few decades. Approximately 18.5% of American children and adolescents are overweight or obese [16]. In Europe, the prevalence of overweight or obese children, combined, varies from approximately 8%-13% in Estonia, France, and Netherlands to 26–32% in Greece, Italy, and Portugal [17,18]. In Finland, it has been estimated that 10% of boys and 18% of girls aged five and 20% of children aged 7–13 are overweight or obese [19,20]. Moreover, a recent study in a population sample of Finnish children aged 2–16 years reported that 26% of boys and 16% of girls were overweight [21].

Previous studies have provided some evidence that a lower intake of polyunsaturated fatty acids (PUFAs) [22,23], a higher intake of saturated fat [24,25], a lower consumption of vegetables and fish [22,23], and a poor adherence to the Mediterranean diet, the Dietary Approaches to Stop Hypertension (DASH) diet, and the Nordic diet [26,27,28,29] may increase the risk of dementia and Alzheimer’s disease among adults and older adults. Similarly, increased cardiometabolic risk has been linked to an increased risk of dementia in older adults [30]. Furthermore, a multidomain intervention including diet, exercise, and cognitive training has been found to improve cognitive functions in elderly individuals at an increased risk of dementia [31]. While a proper diet, sufficient levels of physical activity, and an adequate amount of sleep are essential for normal brain and cognitive development and subsequent academic performance [13,32,33], less is known about the associations of diet, physical activity, and sleep with brain function, cognition, and learning in children and adolescents. 

Due to the increased emphasis on education and learning, evidence on the associations between diet and cognition and academic achievement among children and adolescents would provide valuable information for schools and parents to implement actions to support learning and academic achievement. Therefore, the aim of this review is to provide an overview on the associations of health behaviors, with special reference to diet quality, with brain structure and function, cognition, and academic achievement in children and adolescents under 18 years of age. This literature review is based on the studies conducted from 1998 to 2019 on the associations of health behaviors, including dietary intakes, physical activity, sedentary activities, and sleep with cognition and academic achievements in children and adolescents aged less than 18.

## 2. Results

### 2.1. Health Behaviors and the Brain 

Health-related behaviors such as poor diet quality, physical inactivity, and decreased sleep, and the resulting increased cardiometabolic risk factors, overweight, and obesity, have been related to impaired cognition in children and adolescents. These health-related behaviors are recognized as modifiable risk factors [34]. Here, we elaborate on the existing evidence on the possible effects of physical activity, sedentary behavior, sleep, and diet quality on cognition and the synergistic effects of these lifestyle-related factors on the developing brain and on cognition.

#### 2.1.1. Physical Activity and the Brain

Physical activity has been an integral part of humans’ life and physically active and fit was the basic phenotype of humans for thousands of years [35]. Adequate, safe, and healthy diets may support the levels of physical activity in children and adolescents, while undernourishment may prevent adequate and normal levels of physical activity [36,37]. Physical activity has been associated with increased brain functioning by increasing the blood flow, boosting glucose, and lipid metabolism [38,39,40], and increasing the level of several growth factors such as brain derived neurotrophic factor (BDNF) [39]. Physical activity have also been associated with increasing grey matter volumes in the frontal and hippocampal regions of the brain [40,41]. All these effects of physical activity on brain have been related to improved cognitive functioning [42]. Higher levels physical activity has been linked to better brain functions and structures, cognitive functions, and academic achievement in children and adolescents in some, but not all, cross-sectional studies [33]. While some studies have found an association between objectively measured physical activity, cognition, and academic achievement [43,44,45], others have shown weak, if any, associations of objectively measured physical activity with cognition and academic achievement in children [46,47,48,49] 

Exercise interventions also have provided some evidence that increasing after-school or classroom physical activity improve cognitive functions and academic achievement in children [50,51,52]. Increasing physical exercise levels has been related to structural changes in the brain and subsequently with better cognition and academic achievement in comparison to sedentary individuals [33,53]. Similarly, it has been demonstrated that, compared to sedentary fellows, the children involved in regular aerobic exercise performed better on verbal, perceptual, and arithmetic tests [54,55,56]. However, not all intervention studies have found improved cognitive functions [54] or academic achievement [57]. These results may suggest that physical activity may have weak effects on cognition in mainstream children, but a stronger effect on brain and cognition among children with attention deficit/hyperactivity disorder (ADHD) [58,59], poor academic performance [60], low fitness levels, or who are overweight or obese [61,62,63]. Furthermore, it is possible that not all exercise programs are equally effective in order to improve cognitive functions and academic achievement [64]. 

Coordinative exercise has been found to be more effective than cardiovascular exercise to improve cognitive functions in some studies [37,65]. Furthermore, the effects of physical activity on cognition and academic performance may be context specific; while Mullender-Wijsma et al. [54] found a positive effect of an active academic lessons on academic performance, the same group observed no effect on executive functions among the same children [54]. Some studies have found a direct association of motor skills, but not cardiorespiratory fitness, with academic performance and cognition in children [66,67]. Some evidence also suggests that exercise training augments the effects of DHA on the brain [68]. Furthermore, exercise training has been shown to counteract the adverse effects of unhealthy diet on the brain [68]. 

#### 2.1.2. Sedentary Behavior and the Brain

Sedentary behavior refers to behavior in a sitting or reclined position with low energy expenditure [69]. High levels of screen time, and especially a longer time spent watching television, has been associated with poorer cognitive functions and academic achievement in children and adolescents [70,71,72]. In contrast, some sedentary behaviors, such as reading and drawing, may improve cognitive functions and academic performance in children [73]. These differences in the direction of the associations of different types of sedentary behaviors with cognition and academic achievement may explain the weak and inconsistent associations between objectively measured sedentary time and cognitive functions in youth [44,46,47,74]. The inverse associations of television watching with cognition and academic performance may also be partly due to unhealthy snacking while watching television [75]. Sedentary behavior and particularly television watching and screen-time have been associated with the increased consumption of energy dense food and a lower consumption of fruits and vegetables [75].

#### 2.1.3. Overweight, Obesity, Cardiometabolic Risk and the Brain

Unhealthy diet has been associated with overweight and obesity and increased cardiometabolic risk in children and adolescents [76,77]. Overweight and obesity have been related to poorer cognitive functions [66,78,79,80] and academic achievement [81,82] in children and adolescents. Furthermore, insulin resistance [83] and increased cardiometabolic risk has been linked to decreased hippocampal volume, reduced white matter integrity, frontal lobe white matter volumes in adolescents [84,85], and to poorer cognitive functions in children [86]. Some evidence also suggests an inverse association between systemic inflammation and academic performance in adolescents [87]. 

Cross-sectional studies have found the inverse association between weight status and cognitive performance, especially in tasks involving attention and comprehension [88,89]. Another study comparing overweight, normal weight, and sport trained individuals demonstrated that normal weight individuals had better cognitive functions than overweight individuals [90]. However, the sport trained individuals were better in cognitive performance than normal weight individuals, illustrating that higher cardiorespiratory fitness may play a crucial role in cognitive health [90]. A systematic review of longitudinal studies show inconsistent associations of overweight and obesity with cognition, except in adolescent girls [91]. However, a recent longitudinal study found no causal relationship between childhood obesity and cognitive performance [92]. Nevertheless, the results of one study suggested a positive relationship between body adiposity and academic achievement in children living in India, suggesting that excess energy intake may be better for brain development and learning than undernourishment [78]. 

#### 2.1.4. Sleep and the Brain

Sleep is essential for brain development, learning, and memory [93]. Existing studies have directly linked sleep duration and quality with all three subcomponents of executive function, i.e., working memory, inhibitory control, and cognitive flexibility. A meta-analysis found children’s age as a substantial moderator of the association of sleep duration, quality, and daytime sleepiness with children’s school performance; with younger children showing stronger positive relationship than older children [94]. It is also believed that sleep restriction impairs complex cognitive functions more than the general cognitive abilities, such as language and knowledge in children [95,96]. 

Partial sleep deprivation is associated with impaired working memory in adolescents [97,98,99]. Current evidence demonstrates that decreasing sleep by an hour for four nights deteriorates the working memory of school-aged children [100]. However, increasing sleep duration and quality improved the working memory of adolescents [96]. Some evidence indicates that a short sleeping duration is linked to poorer cognition and academic performance in children and youth [64,101]. Short sleep duration of less than 10 h in early life is associated with impaired task performance in later ages as well—even with normalized sleep duration one year before cognitive evaluation [102]. 

Decreased and disturbed sleep patterns are associated with impaired inhibitory control in children and adults [103,104,105]. Partial sleep deprivation of one to two nights impaired the performance at behavioral tasks and Go/No-Go task, measuring inhibitory control in adults [103,104]. Similarly, moderate sleep deprivation of one week negatively impacted the inhibitory control tasks in school-aged children when compared with those having non-deprived sleep [105]. Moreover, inhibitory control abilities improve by increasing sleep duration in children [96].

Cognitive flexibility is also positively linked with sleep duration in adolescents. An addition of five minutes in sleep duration for two weeks demonstrated a positive impact on the Divided Attention task in chronically sleep deprived adolescents [106]. Single night sleep limitation (5 h) in children aged 10–14 years lead to declined cognitive flexibility and verbal creativity [107], but response inhibition and sustained attention remained unimpaired in children aged 8–15 years [108].

Poor sleep has been linked to an increased risk of obesity among children and adults [109,110,111,112]. There is also some evidence that short sleep length is associated with unhealthy diet and eating patterns that favor energy dense foods, such as fast food and candy, in children and adolescents [113,114]. A recent study found that shorter sleep duration (<8 h) in adolescents is related to an increased consumption of fats and a decreased intake of carbohydrates that may predispose them to an increased risk of obesity [115]. Overall, most of the observational and experimental studies have found cross-sectional associations between short sleep duration (<8 h per night) and increased body mass index or obesity, impaired glucose metabolism, and dysregulation of appetite, but the longitudinal evidence to establish the causal relationship is still limited [116].

### 2.2. Nutrition and the Brain

Nutrition plays crucial role throughout life in the development and protection of the brain and cognition [117]. Diet quality is important from childhood to adolescence for various brain functions such as neurogeneration, axonal and dendritic growth, synaptic formation, and the myelination of axons [36]. Inadequate and low intake of energy, protein, fatty acids, and micronutrients disrupt these neurodevelopmental processes [36]. Here, we elaborate on the association of nutrients, food, and dietary patterns with cognition and academic achievement in children and adolescents.

#### 2.2.1. Nutrients, Cognition, and Academic Achievement

The evidence from previous studies suggest that a higher or adequate intake of some nutrients, such as polyunsaturated fatty acids (PUFAs), and a lower intake of saturated fatty acids are related to better cognitive functions and academic achievement in children [118,119,120].

A higher dietary intake of PUFAs has been related to better short-term memory in children [120,121,122]. The results of one study suggested that replacing carbohydrates or saturated fat with PUFAs was associated with better short term memory [122]. Few studies have investigated the associations of the plasma biomarkers of PUFA status with cognition in children and these studies have provided inconsistent results [122,123]. Montgomery et al. [123] found no association between the proportion of docosahexaenoic acid (DHA) in blood and working memory, but observed a direct relationship between the proportion of combined DHA and eicosapentaenoic acid (EPA) in blood and working memory in children. Boucher et al. [122] found no associations between the proportions of DHA, EPA, or other n-3 fatty acids in blood phospholipids and working memory in children. One study found that higher plasma proportions of EPA and DHA and EPA to the Arachidonic acid ratio were associated with better reasoning skills in overweight and obese children, but not in normal weight children [124]. Furthermore, one study found that increased whole-blood DHA and EPA concentrations after a three month school meal intervention were related to greater improvements in reading performance in 9 to 11-year old children [125]. However, the results of the Special Turku Coronary Risk Factor Intervention Project (STRIP) Study suggested that a limiting dietary fat intake, while increasing the proportion of PUFA and monounsaturated fatty acids, had no effect on language skills in young children [126]. 

Studies have suggested that a higher intake of n-3 PUFA supplements, especially DHA, may improve cognitive functions during the perinatal period and infancy, but not in childhood or adulthood [118,122,127,128]. Recently, Johnson et al. [129], however, found that the increased intake of omega 3 and 6 fatty acids (DHA, EPA, and gamma-linolenic acid) for three months had a positive effect on prerequisites of reading and that the effects were particularly strong among those with a poorer attention. 

Zhang et al. [120] found an inverse association of dietary cholesterol intake with working memory among 6–16-year-old children and adolescents. Baym et al. [119] observed inverse associations of the consumption of saturated and trans-fats with working memory in school-aged children aged 7–9.

The chronically increased consumption of fructose may have deleterious effects on the brain [130]. Fructose ingestion has been observed to decrease blood flow to the hippocampus in mice [131]. Increased blood flow and blood volume in the hippocampus has been linked to increased neurogenesis in the dentate gyrus [132]. Furthermore, high fructose intake has been found to exacerbate the negative effects of DHA deficient diets on synaptic plasticity in a rodent model [133]. A diet that is high in saturated fat and fructose has been found to decrease insulin signaling in the hippocampus and decrease hippocampal weight, impair the dendritic tree, decrease synaptophysin content, and increase tau phosphorylation [134]. A regular consumption of sucrose sweetened beverages may also increase amyloid-β levels and accumulation in the brain [135].

A higher intake of total fiber and insoluble fiber has been linked to better cognitive control in children [128]. Furthermore, breakfast with a lower glycemic load acutely improved working memory in 5–11-year-old children [136].

A protein and energy rich formula during infancy had a positive association with caudate nucleus volume and verbal IQ at the age of 10, than those who were fed with the standard formula among pre-term children [137]. 

In addition to macronutrients, micronutrients are also important for brain development. Inadequate intake of micronutrients such as iron, iodine, zinc, and vitamins B, D, and E, have been found to impair neuronal proliferation, axonal and dendritic growth, synaptic formation, pruning, and function, and axonal myelination [36,138,139]. A systemic review of randomized controlled trials among children aged 4–18 demonstrated an improvement in fluid intelligence after micronutrient supplementation, especially among those having iron or iodine-deficiency at baseline, but the results among healthy subjects without nutritional deficiencies remain inconsistent [140].

#### 2.2.2. Foods, Cognition, and Academic Achievement

Food is the combination of various nutrients, therefore, undernourishment and low availability of nutritionally adequate and safe foods has been observed to decrease cognitive functions in children [141,142]. Higher consumption fish has been linked to better cognition and academic achievement in children and adolescents [143,144,145]. In a Norwegian study, a higher consumption of fruit, especially berries, was related to better academic achievement in female and male adolescents, while a higher consumption of vegetables was only related to better academic performance in adolescent girls aged 15–17 years [146]. Some other studies have also reported a direct relationship between vegetable consumption and academic performance [144,147]. Furthermore, low consumption of fruit, berries, vegetables, and high-fiber grain products and high consumption of red meat have been associated with poorer cognition in children aged 6–8 years [148].

Finally, the results of some studies suggest that a higher consumption of soft drinks and sweet beverages is related to poorer academic achievement [144]. 

#### 2.2.3. Dietary Patterns, Diet Quality Indices, Cognition, and Academic Achievement

Dietary patterns and diet quality indices have been suggested to better reflect a real-life diet because nutrients do not naturally exist in isolation. Food is the combination of various nutrients that act synergistically and are interrelated [29,149]. There are various dietary scores and indices used to assess the dietary patterns and diet quality, for example, Mediterranean dietary score, Baltic Sea Diet Score (BSDS), Dietary Approaches to Stop Hypertension (DASH) score, Finnish Children Healthy Eating Index (FCHEI), and the Healthy Eating Index.

While evidence on the associations of dietary patterns with academic achievement is limited, some studies suggest a direct relationship of adherence to the Mediterranean style diet assessed by the KIDMED index [150,151] and the Healthy Eating Index [147] with academic achievement in children and adolescents. The Healthy Eating Index 2005 has been linked to better cognitive control in 7–9-year-old children [128]. A higher BSDS reflecting a better adherence to healthy diet in the Nordic countries and the better adherence to the DASH diet have been linked to better cognition in children and particularly in boys aged 6–8 [143]. 

Recently, a healthier diet assessed by the BSDS and the FCHEI in Grade 1 was related to better reading skills in Grades 1–3 in Finnish children [148]. However, a Mediterranean Diet Score reflecting Mediterranean style diet was not associated with academic performance [148].

A three month randomized controlled school lunch trial showed that healthy Nordic diet improved reading skills, but impaired attention in children aged 10 years [152]. However, in contrast to some previous studies, the effects of the intervention were stronger in boys, in those from families with more educated parents, and in those with normal/good baseline reading skills [153].

A dietary pattern high in fruits, vegetables, and home-prepared foods at the age of 6 and 12 months has been associated with better cognitive function at the age of four [154]. Fruits, especially berries, vegetables, green tea, red wine, and chocolate are rich in flavonoids. A higher intake of flavonoids has been associated with better cognition [32]. Flavonoids are polyphenols and have been suggested to protect neurons against neurotoxin-induced injuries, decrease neuroinflammation, and increase neuroprotective signaling [32,138]. 

Dietary patterns, including a high consumption of sausage, fast food, snacks, and sugar sweetened beverages at the age of three years, has been related to poorer academic achievement at the age of 10 years among children [155]. Western dietary patterns that are high in take-away foods, red and processed meat, soft drinks, and fried and refined food has been linked to poorer cognition at the age of 17 [156]. 

## 3. Discussion

The current review provides an elaborated overview of existing literature on the associations of health behaviors, including diet quality and dietary intakes, physical activity, sedentary activities, and sleep with brain structure and function, cognition, and academic achievement in children and adolescents under 18 years of age, with a special reference to diet quality. However, this review is not systematic and all the relevant studies were not described. Therefore, this review should not be treated as the exhausting evidence synthesis. Instead, we aimed to provide generalized view of available literature on the topic. Furthermore, we did not review the associations of other health behaviors, such as smoking or alcohol or substance abuse, with brain structure and function, cognition, and academic achievement. Therefore, we cannot exclude the possibility that they effect on the results of reviewed studies. 

### 3.1. Health Behaviors and Cogntion

Despite the increased interest, the evidence on the associations of nutritional factors, physical activity, sedentary activities, and sleep with brain structure and function, cognition, and academic achievement in children and adolescents is still mainly based on cross-sectional studies with mixed results.

Some cross-sectional studies have found direct associations of physical activity with cognitive functions, including learning and academic outcomes, and neuroelectric processing in children [157,158]. Other studies have found no relationship or they have found inverse associations between physical activity, physical fitness, and cognitive functions in children and adolescents [47,48,159]. A few interventional studies with varying study protocols have provided some evidence that increasing after-school or classroom physical activity may improve cognitive functions, neural processing, and academic achievement in children [33,54,63,158,160]. However, some studies also suggest that increasing the time allocated to physical activity has no effect on cognition or academic achievement [41,52,57]. Higher physical fitness has been linked to better cognition and academic achievement in children [33]. Prospective investigations on the associations between physical fitness, cognition, and academic achievement are few, but the evidence from these studies indicate that children and adolescents who maintain high levels of physical fitness have better cognitive and academic performance than those who have consistently low levels of physical fitness [33,161,162,163]. These studies have shown that changes in physical fitness during follow-up have only weak associations with the changes in cognitive functions or academic achievement [161,162,163]. Furthermore, most studies have investigated the associations of cardiorespiratory fitness with cognition and academic achievement [33], although the results of some studies indicate that motor skill training and motor skills have a stronger relationship to cognition and academic achievement than cardiorespiratory fitness in children [65,67].

Sedentary behaviors, such as reading and drawing, may be associated with improved cognitive functions and academic performance in children [74]. In contrast, sedentary behaviors with high levels of screen time have been associated with poorer cognitive functions and academic achievement in children and adolescents [70,71,72]. The inverse associations of screen time with cognition and academic performance may also be partly due to the unhealthy snacking on energy dense food and a lower consumption of fruit and vegetables while watching television [75].

The sleep duration and quality have directly linked with all three subcomponents of executive function, i.e., working memory, inhibitory control, and cognitive flexibility. Children’s age has been found to be a substantial moderator of the association of sleep duration and quality with children’s school performance; younger children showed a stronger relationship than older children [94]. It is also believed that sleep restriction impairs the complex cognitive functions more than the general cognitive abilities such as language and knowledge in children [95,96]. Poor sleep and short sleep length is associated with unhealthy diets and eating patterns favoring energy dense foods, such as fast food and candy, in children and adolescents [113,114]. Overall, most of the observational and experimental studies have found cross-sectional associations between short sleep duration (<8 h per night) and increased body mass index or obesity, impaired glucose metabolism, and dysregulation of appetite, but the longitudinal evidence to establish the causal relationship is still limited [117].

### 3.2. Nutrition and Diet Quality.

Diet may have direct and indirect effects on the brain (Figure 1). However, the direct and indirect effects of diet on the brain are often overlapping and synergistic [29,83]. Undernourishment and the low availability of nutritionally adequate and safe foods have been observed to decrease cognitive functions in children [141,142].

The evidence suggests that the low intake of PUFAs, especially DHA and EPA during early years, may attenuate brain and cognitive development. A low consumption of fish, fruit and berries, and vegetables, and a high consumption of fast foods, have been linked to poorer cognitive functions and academic achievement in children and adolescents. The results of some studies also suggest that dietary patterns high in vegetables and fruits are directly and dietary patterns high in fast food, sausages, and soft-drinks are inversely associated with cognition. Furthermore, diet quality indices such as different Mediterranean diet scores, the DASH diet score, the Healthy Eating Index, and the Baltic Sea Diet Score have been directly related to cognition and academic achievement. There are also some but still very limited evidence on the associations of high intake of saturated fat and low intake of fiber and low consumption of high-fiber grain products and high consumption of red meat with cognition in children and adolescents. There are promising results that increasing DHA and EPA intake and improving a quality of school lunches may improve cognition and academic performance, but the evidence is limited. These are summarized in Table 1.

While the evidence on the associations of PUFA with cognition and academic achievement in children and adolescents is still limited, the positive association between PUFA and brain health is physiologically plausible. PUFAs are essential for normal brain development [164] as they regulate cell membrane dynamics and modulate the endocannabinoid system that, in turn, regulates neurotransmission and participates in synaptic plasticity [164]. n-3 PUFA DHA has been linked to neurogenesis and neural survival [164]. A higher intake of DHA may also reduce apoptotic signaling in the brain and increase brain derived neurotrophic factor (BDNF) concentration [164]. BDNF has been found to improve neural survival, neural maintenance, neurogenesis, synaptic plasticity, long-term potentiation, and neurotransmitter release [68,132,138]. Furthermore, the results of some studies suggest an inverse association between the dietary intake of saturated fat and BDNF [165]. Similarly, a diet high in saturated fat may have adverse effects on synapsin I, growth associated protein 43 (GAP-43), and cyclic AMP response element-binding protein (CREB), which have been linked to synaptic plasticity, neurotransmitter release, and axonal growth [165].

A Western diet usually denotes a diet high in saturated fat and refined carbohydrates [83,165]. Most studies on the effects that Western diets have manipulated the consumption of saturated fat and refined carbohydrates [83]. A diet high in saturated fatty acids and refined carbohydrates has been linked to increased oxidative stress and a subsequent decrease in hippocampal BDNF, and impaired BDNF-related synaptic plasticity [68,165]. A Western diet may impair brain health and cognition directly or indirectly. A diet high in saturated fat and refined carbohydrates may cause insulin resistance, reduce BDNF and other neurotrophin concentrations, and thereby impair synaptic plasticity and neurogenesis in the hippocampus, increase neuroinflammation, and alter the blood–brain barrier [83,165]. Alterations in the blood–brain barrier permeability may increase the permeability of toxins and β-amyloid peptides and thereby impair brain health and cognitive functions [86].

Dietary patterns and diet quality indices have been suggested to better reflect a real life diet and the improvements in dietary patterns may be more easily translated to real life conditions than those in single nutrients, because single nutrients do not normally exist in isolation and are interrelated and synergistic [29,149]. There are also differences in food cultures and food choices between geographical regions and populations. While the Mediterranean style diet has frequently been used to describe healthy diets [29], it may not be easily translated to other countries, such as the Nordic countries. However, there seems to be some specific nutrients that have a pronounce effect on the brain and cognition. For example, PUFAs has been found to be essential for normal brain development [164] and a recent study suggested that increased DHA and EPA levels explained approximately 20% of the effects of school-lunch intervention on academic performance in children [152]. However, the results of some cross-sectional studies are not as straightforward. Khan et al. [128] found a positive association between intake of dietary fiber and response accuracy in the Flanker task, but no association of intake of dietary fiber and the ability to sustain response accuracy between two task requiring different amounts of cognitive control. However, they found that the Healthy Eating Index 2015 was inversely associated with the interference score, suggesting that children with a healthier diet were better able to sustain the response accuracy with increasing cognitive demands. Another study found that low consumption of fruit, berries, vegetables, and high-fiber grain products and high consumption of red meat have been associated with poorer cognition in children aged 6–8-years [143]. Nevertheless, Baltic Sea Diet Score and the DASH score had stronger associations with cognition than any single food item [143].

Better adherence to different diet quality indices and healthier dietary patters have been associated with better cognitive functions and academic achievement [128,143,147,148,151]. These diet quality indices mostly describe a diet high in vegetables, fruit and berries, high fiber grain products, and fish and low in fast foods, saturated fat, and refined carbohydrates. However, some studies have observed that that the magnitude of the associations of different diet quality indices with cognition and academic performance are not equal. One study found that while the Baltic Sea Diet Score and Finnish Children Healthy Eating Index were directly related to academic performance in Finnish children aged 6–8 years, Mediterranean Diet Score was not related to academic performance [148]. The authors speculated that these differences in the strength of the associations might be explained by the low variance in the Mediterranean diet score or differences in the components emphasized. For example, while the Baltic Sea Diet Score and Finnish Children Healthy Eating Index emphasize the intake of polyunsaturated fatty acids and the consumption low-fat milk, whereas the Mediterranean diet score gives emphasis to the ratio of monounsaturated to saturated fatty acids and low consumption of milk. Milk is commonly fortified with vitamin D in Finland, and it is the major dietary source of vitamin D among Finnish children. Furthermore, another study found a stronger association of Baltic Sea Diet Score than the DASH score with cognition [143]. These results may also indicate that diet quality indices developed for specific geographical locations and by using some specific components may better reflect a healthy diet than diet quality indices developed for other cultures. Nevertheless, the evidence from these studies suggest that a diet high in vegetables, fruit and berries, and fish are linked to better cognition and academic performance in children and adolescents.

## 4. Conclusions

The aim of this review was to provide an overview on the associations of health behaviors with brain structure and function, cognition, and academic achievement in children and adolescents with a special reference to diet quality. Overall, there is some evidence that in addition to poor diet quality, excess adiposity, increased cardiometabolic risk, physical inactivity, high levels of screen-time, poor physical fitness, and low levels of sleep may impair cognitive functions and academic performance in children and adolescents. While the available evidence and physiological mechanisms suggest that diet may have direct, indirect, and synergistic effects on the brain and cognition with physical activity, sedentary behaviors, cardiometabolic health, and sleep, the associations are modest. The evidence also suggests that diet and physical activity have at least partly different pathways [68]. Accordingly, some studies have found that a healthier diet is related to better cognition and academic performance, independent of physical activity, sedentary behaviors, or physical fitness [143,148]. While they seem to be related to cognition, independent of each other, the magnitude of the effects has been modest. Therefore, it is possible that integrating healthy diet, physically active lifestyle, and adequate levels of sleep would provide optimal circumstances for brain development and learning during childhood and adolescence. However, the evidence is mostly based on cross-sectional data, and there is need for longitudinal and intervention studies on the effects of diet, physical activity, sedentary behavior, and sleep on cognition and academic performance in children and adolescents and to expose the complex interactions among these health related behaviors.

## Figures and Tables

**Figure 1 ijerph-17-00953-f001:**
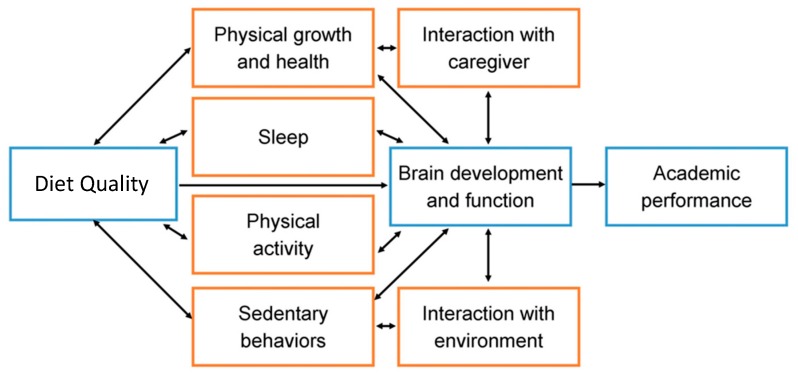
Hypothetical and simplified model on the direct and indirect effects of nutritional status and diet on brain development and academic performance in children (adapted from Reference [36]).

**Table 1 ijerph-17-00953-t001:** Associations of dietary factors with cognition and academic performance.

Dietary Factors	Evidence in Relation Cognitive Functions
**Nutrients, cognition and academic achievement**	
Iron	Iron deficiency leading to anemia may negatively impact on brain and cognitive development
Docosahexaenoic acid and eicosapentaenoic acid	Low intake during early years may attenuate brain and cognitive development
**Foods, cognition and academic achievement**	
Fish	A higher consumption of fish has been associated with better cognitive functions
Fruit, berries, and vegetables	A higher consumption of fruit, berries, and vegetables have been associated with better cognitive functions
Fast foods	A higher consumption of fast foods has been associated with poorer cognitive functions
**Dietary patterns, diet quality indices, cognition and academic achievement**	
Dietary patterns	Dietary patterns high in vegetables, fruits, and home-prepared foods have been associated with better cognitive functions. Dietary patterns high in fast foods, red meat, soft-drinks, and fried and refined foods have been linked to poorer cognitive functions
Diet quality indices (DASH, BSDS, FCHEI, KIDMEX, HEI-2005)	A better adherence to pre-specified diets (i.e., higher scores) has been associated with better cognitive functions and academic performance

Abbreviations: DASH, Dietary Approaches to Stop Hypertension; BSDS, Baltic Sea Diet Score; FCHEI, Finnish Children Healthy Eating Index; HEI-2005, Healthy Eating Index 2005.

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
