# Peer review of "An Overview on the Associations between Health Behaviors and Brain Health in Children and Adolescents with Special Reference to Diet Quality"

_ijerph, 2020, doi:10.3390/ijerph17030953_

Round 1

Reviewer 1 Report

Looking much better. Minor changes.

Thank you to the authors for making a considered effort to take our comments on board. The organization of the manuscript is much improved, and the Discussion section especially is more thoughtful. It is a challenge to bring together so many factors in one review and I commend you on your efforts.

Here are my comments:

Line 242: This line about micronutrients, while interesting, comes out of nowhere. Consider a better way to integrate this information.

Line 379: you discuss Western diet and cognition too briefly here - there is growing literature in human adults about the negative impacts of cognitive function from cross-sectional (Francis & Stevenson, 2011) and experimental studies (Attuquayefio et al. 2016; Stevenson et al 2020). It is also associated with lower hippocampal volume (Jacka et al. 2015). Kanoski, Davidson and Hsu have done a lot of work on rodents looking at the brain effects too. 

General comment: many type of cognitive functions appear to be affected by diet, sleep, physical activity etc. Is there any evidence that one type of cognitive function is more/less affected by each of these factors? This would be interesting to know, since they may have different effects (perhaps via different mechanisms). Or is there no evidence of this? Some consideration is warranted

Reviewer 2 Report

Dear authors;

The article is fine and provides a good introduction into the state of science.

However, I would like to point out some comments/suggestions I think could improve the MS.

First of all, some points have been omitted from the analysis (i.e. alcohol, smoking habit or (why not?) drug addiction or substance abuse, unfortunately common in teenagers).

In addition, I miss a deeper discussion and the introduction of own points of view. Authors should go to great lengths and provides argumentes that support or not facts cited in the papers employed.

 Finally, the conclusion maybe should be more precise and concise, reflecting the step forward the MS achives.

Author Response

This manuscript is a resubmission of an earlier submission. The following is a list of the peer review reports and author responses from that submission.

Round 1

Reviewer 1 Report

This review paper provided an overview on the association of healthy lifestyle with brain, cognition, and academic performance in children and adolescents with reference to diet quality, physical activity, appropriate screen-time, and adequate sleep. This paper cited adequate amount of references related with this topic, and provided a comprehensive review. However, as the authors also mentioned, much of the evidence was based on cross-sectional data, not intervention studies. Therefore, the significance of this review is not considered high. But overall, there is still merit and contribution of this paper to the field.

Reviewer 2 Report

The subject is very interesting. However, as the manuscript is not well organised and lacks thorough discussion. A thorough revision is required in all sections.

The second part of the title must be changed (An Overview with Special Reference to Diet Quality). As stands implies the focus is on diet, while in lines 25-27 and 73-76, various lifestyle-related factors, not only diet, have been explained. OR if this is not the case the conclusion should be changed.

Line 14, “children and adolescents”; please add the age range

Line 18, berries are a type of fruit.

Line 21-24: Please rewrite the sentence. As stands is not clear where the yellow highlighted section belong to?

“Diet may have direct, indirect and synergistic effects on brain and cognition with physical activity, sedentary behaviours, cardiometabolic health and sleep.”

Line 24: “Although they seem to be related… “: what does “they” refer to?

Line 37: please reword: “Given that poor nutrition is important for brain development, cognition …”. How poor nutrition can help brain development?

Line 37-39: the whole sentence should be re-written. It is not easy to read and link two sections together.

Line 40-45: the reported statistics are fine. But how are they related to the subject of this paper? This needs to be mentioned. As stands, it is just reporting on the unhealthy habits of children and adolescents, without pointing to their potential relationship with the brain, cognition, etc. etc.

Same comment as above for line 46-52.

Line 71: obesity, increased cardiometabolic risk are not health behaviours, they are consequences of behaviours. Please revise.

Line 73: what are recognized as modifiable risk factors?

Line 77 and Figure 1: This is not correct. Nutritional status is something totally different from the diet. Nutritional status is the condition of the body following dietary intake. What is shown in Figure 1 it is a totally different subject compared with the relationship between diet and brain. This must be removed/changed. Or at least the way it has been modified or mentioned is not correct.

Section nutrition and diet quality: should be organized. Authors can first discuss the relationship with nutrients and then talk about dietary quality and dietary patterns. Currently, authors have discussed fat and sugar under nutrients and only western diet as a dietary pattern. More comprehensive analysis of the available literature is needed.

Line 122-132. This section is very general. More explanation is required. How these are related to brain health, etc.

Line 133-170. This section is not well organized and explained. Physical activity is not the same as physical fitness. Sub-sections need to be created and each subject must be discussed thoroughly and in details. No reason/explanation for such relationship has been provided. What mechanisms are involved?

Line 244: what does food mean in this context? This section must be under section 5, dietary patterns.

Merge Sections 3, 4, 5 with Nutrition and diet quality section (line 77). And discuss them in subsections.

The conclusion is very long. Please revise and make it to the point.

Reviewer 3 Report

It is a review to address the effects of diet on the cognitive development among children and adolescents. Although a lot of evidence were cited the way it was done is very confusing. I would suggest the authors review the writing because some phrases are not clear. The title should be reviewed. I would suggest rethinking the proposal by choosing only one relevant aspect and go deep on it. The authors mixed too many variables and aspects that perhaps do not fit as the activity level or obesity. Figure 1 is not clear because what does it mean "nutritional status" and where is the "diet" as a variable? Table 1 is not correct since it is more a conclusion of the authors than evidence which should address the references. The Conclusion, item 6, should be reviewed since its content does not look like a conclusion but a continuation or a reinforcement of what was previously discussed. 

Reviewer 4 Report

This narrative review brings together the complex factors that impact cognition/academic achievement in children and adolescents, with a focus on diet quality and nutrients. This is no small task, and I commend the authors on bringing this important information together. While reading about the independent effects of each of the listed factors was interesting, comprehensive and needed, I was very much left with the sense of still not knowing or understanding how these factors interact (or the degree to which each one impacts cognition). My major concern is that the review has not fulfilled its own purpose. So, while the independent effects on cognition are known, their interactions are not well understood. I would imagine that at least one of the listed references also investigated whether physical activity/sedentary behaviour/sleep/obesity affected cognition and academic performance in children. It would be extremely valuable to the reader and the scientific community at large if you were able to tease apart these complex interactions. For instance, if adiposity and diet quality affect cognitive performance while sleep quality does not, it might suggest that the former are more impactful than the latter. As I mentioned earlier, a review of this nature is no small task but would be invaluable. This would be a more comprehensive narrative review than its current state. If teasing apart these complex interactions is too difficult or too speculative in human studies, then this point needs to be made in the manuscript.

My minor comments are:

Line 40: "grow" - this word could be changed to something else, given the context of adolescents growing developmentally.

Line 49: Extra "the" in front of Estonia

Line 58: What is meant by 'multidomain' diet?

Lines 62-63: Again, here you mention the associations between factors, but this question still feels unanswered by the end

Line 71: Obesity and cardiovascular risk, while related to impaired cognition, are themselves not strictly 'health behaviours' . Consider revising this sentence

Line 79: these findings are mainly in animals - what about humans? For instance, have a look at Shi et al (2019) Iron-related dietary pattern increases the risk of poor cognition [ https://doi.org/10.1186/s12937-019-0476-9]. Reference 36 provides information mainly based on rodent work, while 37 is based around humans. This point needs to be clarified

Lines 88-107: These findings are in rodents and this point should be clarified

Line 102: "dentate gurys" should be "dentate gyrus"

Line 175: What is the direction of this stronger relationship?

Lines 211-232: Dietary/supplement intake of PUFAs seems to be beneficial, yet biomarkers do not reflect this pattern of findings. Could the authors comment on/discuss this more?

Line 277: Your conclusions begin with statements around iron deficiency, when you have made only 1 mention of iron previously. This conclusions seems unwarranted given the strength of evidence on other dietary factors. 

Line 277: There has been no mention of anemia, nor its implications in cognition. Consider removing it.

Line 291: Table 1 subheadings could be simplified to "Nutrients" , "Foods" and "Dietary pattern and quality indices" for readability

Lines 306-309: This distinction between nutrients, food and dietary pattern is important and should probably go earlier in the manuscript to justify the earlier headings/subheadings.